# Sentence Pair Scoring: Towards Unified Framework for Text Comprehension

**redacted**

## Abstract

We review the task of Sentence Pair Scoring, popular in the literature in various forms — viewed as Answer Sentence Selection, Semantic Text Scoring, Next Utterance Ranking, Recognizing Textual Entailment, Paraphrasing or e.g. a component of Memory Networks.

We argue that all such tasks are similar from the model perspective and propose new baselines by comparing the performance of common IR metrics and popular convolutional, recurrent and attention-based neural models across many Sentence Pair Scoring tasks and datasets. We discuss the problem of evaluating randomized models, propose a statistically grounded methodology, and attempt to improve comparisons by releasing new datasets that are much harder than some of the currently used well explored benchmarks. We introduce a unified open source software framework with easily pluggable models and tasks, which enables us to experiment with multi-task reusability of trained sentence models.

## 1 Introduction

An NLP machine learning task often involves classifying a sequence of tokens such as a sentence or a document, i.e. approximating a function $f_1(s) \in [0, 1]$ (where $f_1$ may determine a domain, sentiment, etc.). But there is a large class of problems that involve classifying a pair of sentences, $f_2(s_0, s_1) \in \mathbb{R}$ (where $s_0$, $s_1$ are sequences of tokens, typically sentences).

Typically, the function $f_2$ represents some sort of *semantic similarity*, that is whether (or how much) the two sequences are semantically related.

This formulation allows $f_2$ to be a measure for tasks as different as topic relatedness, paraphrasing, degree of entailment, a pointwise ranking task for answer-bearing sentences or next utterance classification.

In this work, we adopt the working assumption that there exist certain universal $f_2$ type measures that may be successfuly applied to a wide variety of semantic similarity tasks — in the case of neural network models trained to represent universal semantic comprehension of sentences and adapted to the given task by just fine-tuning or adapting the output neural layer (in terms of architecture or just weights). Our argument for preferring $f_2$ to $f_1$ in this pursuit is the fact that the other sentence in the pair is essentially a very complex label when training the sequence model, which can therefore discern semantically rich structures and dependencies. Determining and demonstrating such universal semantic comprehension models across multiple tasks remains a few steps ahead, since the research landscape is fragmented in this regard. Model research is typically reported within the context of just a single $f_2$-type task, each dataset requires sometimes substantial engineering work before measurements are possible, and results are reported in ways that make meaningful model comparisons problematic.

Our main aims are as follows. (A) Unify research within a single framework that employs task-independent models and task-specific adaptation modules. (B) Improve the methodology of model evaluation in terms of statistics, comparing with strong non-neural IR baselines, and introducing new datasets with better characteristics. (C) Demonstrate the feasibility of pursuing universal, task-independent $f_2$ models, showing that even simple neural models learn universal semantic comprehension by employing cross-task transfer learning.

The paper is structured as follows. In Sec. 2, we outline possible specific $f_2$ tasks and available datasets; in Sec. 3, we survey the popular non-neural and neural baselines in the context of these tasks; finally, in Sec. 4, we present model-task evaluations within a unified framework to establish the watermark for future research as well as gain insight into the suitability of models across a variety of tasks. In Sec. 5, we demonstrate that transfer learning across tasks is helpful to powerfully seed models. We conclude with Sec. 6, summarizing our findings and outlining several future research directions.

## 2 Tasks and Datasets

The tasks we are aware of that can be phrased as $f_2$-type problems are listed below. In general, we primarily focus on tasks that have reasonably large and realistically complex datasets freely available. On the contrary, we have explicitly avoided datasets that have licence restrictions on availability or commercial usage.

### 2.1 Answer Sentence Selection

Given a factoid question and a set of candidate answer-bearing sentences in encyclopedic style, the first task is to rank higher sentences that are more likely to contain the answer to the question. As it is fundamentally an Information Retrieval task in nature, the model performance is commonly evaluated in terms of Mean Average Precision (MAP) and Mean Reciprocial Rank (MRR).

This task is popular in the NLP research community thanks to the dataset introduced in (Wang et al., 2007) (which we refer to as `wang`), with six papers published between February 2015 and 2016 alone and neural models substantially improving over classical approaches based primarily on parse tree edits.[1] It is possibly the main research testbed for $f_2$-style task models. This task has also immediate applications e.g. in Question Answering systems.

In the context of practical applications, the so-far standard `wang` dataset has several downsides we observed when tuning and evaluating our models, illustrated numerically in Fig. 1 — the set of candidate sentences is often very small and quite uneven (which also makes rank-based measures

unstable) and the total number of individual sentence pairs as well as questions is relatively small. Furthermore, the validation and test set are very small, which makes for noisy performance measurements; the splits also seem quite different in the nature of questions since we see minimum correlation between performance on the validation and test sets, which calls the parameter tuning procedures and epoch selection for early stopping into question. Alternative datasets WikiQA (Yang et al., 2015) and InsuranceQA (Tan et al., 2015) were proposed, but are encumbered by licence restrictions. Furthermore, we speculate that they may suffer from many of the problems above[2] (even if they are somewhat larger).

To alleviate the problems listed above, we are introducing a new dataset `yodaqa/large2470` based on an extension of the `curatedv2` question dataset (introduced in (Baudiš and Šedivý, 2015), further denoisified by Mechanical Turkers) with candidate sentences as retrieved by the YodaQA question answering system (Baudiš, 2015) from English Wikipedia and labelled by matching the gold standard answers in the passages.[3]

Motivated by another problem related to the YodaQA system, we also introduce another dataset `wqmprop`, where $s_0$ are again question sentences, but $s_1$ are English labels of properties that make a path within the Freebase knowledge base that connects an entity linked in the question to the correct answer. This task (**Property Selection**) can be evaluated identically to the previous task, and solutions often involving Convolutional Neural Networks have been studied in the Question Answering literature (Yih et al., 2015) (Xu et al., 2016). Our sentences have been derived from the WebQuestions dataset (Berant et al., 2013) extended with the moviesE dataset questions (originally introduced in (Baudiš and Šedivý, 2015)); the property paths are based on the Freebase knowledge graph dump, generated based on entity linking and exploration procedure of YodaQA v1.5.[4]

Fig. 1 compares the critical characteristics of

---

[1] http://aclweb.org/aclwiki/index.php?title=Question_Answering_(State_of_the_art)

[2] Moreover, InsuranceQA is effectively a classification task rather than a ranking task, which we do not find as appealing in the context of practical applications.

[3] Note that the `wang` and `yodaqa` datasets however share a common ancestry regarding the set of questions and there may be some overlaps, even across train and test splits. Therefore, mixing training and evaluation on wang and yodaqa datasets within a single model instance is not advisable.

[4] https://github.com/brmson/dataset-factoid-webquestions branch *movies*

the datasets. Furthermore, as apparent below, the baseline performances on the newly proposed datasets are much lower, which suggests that future model improvements will be more apparent in evaluation.

## 2.2 Next Utterance Ranking

(Lowe et al., 2015) proposed the new large-scale real-world Ubuntu Dialogue dataset for an $f_2$-style task of ranking candidates for the next utterance in a chat dialog, given the dialog context. The technical formulation of the task is the same as for Answer Sentence Selection, but semantically, choosing the best followup has different concerns than choosing an answer-bearing sentence. Recall at top-ranked 1, 2 or 5 utterances out of either 2 or 10 candidates is reported; we also propose reporting the utterance MRR as a more aggregate measure. The newly proposed Ubuntu Dialogue dataset is based on IRC chat logs of the Ubuntu community technical support channels and contains casually typed interactions regarding computer-related problems.[5] While the training set consists of individual labelled pairs, during evaluation 10 followups to given message(s) are ranked. The sequences might be over 200 tokens long.

Our primary motivation for using this dataset is its size. The numerical characteristics of this dataset are shown in Table 1.[6] We use the v2 version of the dataset.[7] Research published on this dataset so far relies on simple neural models. (Lowe et al., 2015) (Kadlec et al., 2015)

## 2.3 Recognizing Textual Entailment and Semantic Textual Similarity

One of the classic tasks at the boundary of Natural Language Processing and Artificial Intelligence is the inference problem of Recognizing Textual Entailment (Dagan et al., 2006) — given a pair of a factual sentence and a hypothesis sentence, we are to determine whether the hypothesis represents a contradiction, entailment or is neutral (cannot be proven or disproven).

We include two current popular machine learning datasets for this task. The Stanford Natural Language Inference **SNLI** dataset (Bowman et al.,

2015) consists of 570k English sentence pairs with the facts based on image captions, and 10k + 10k of the pairs held out as validation and test sets. The **SICK-2014** dataset (Marelli et al., 2014) was introduced as Task 1 of the SemEval 2014 conference and in contrast to SNLI, it is geared at specifically benchmarking semantic compositional methods, aiming to capture only similarities on purely language and common knowledge level, without relying on domain knowledge, and there are no named entities or multi-word idioms; it consists of 4500 training pairs, 500 validation pairs and 4927 testing pairs.

For the SICK-2014 dataset, we also report results on the Semantic Textual Similarity. This task originates in the STS track of the SemEval conferences (Agirre et al., 2015) and involves scoring pairs of sentences from 0 to 5 with the objective of maximizing correlation (Pearson's $r$) with manually annotated gold standard.

## 3 Models

As our goal is a universal text comprehension model, we focus on neural network models architecture-wise. We assume that the sequence is transformed using $N$-dimensional word embeddings on input, and employ models that produce a pair of sentence embeddings $E_0$, $E_1$ from the sequences of word embeddings $e_0$, $e_1$. Unless noted otherwise, a Siamese architecture is used that shares weights among both sentenes.

A scorer module that compares the $E_0, E_1$ sentence embeddings to produce a scalar result is connected to the model; for specific task-model configurations, we use either the **dot-product** module $E_0 \cdot E_1^T$ (representing non-normalized vector angle, as in e.g. (Yu et al., 2014) or (Weston et al., 2014)) or the **MLP** module that takes element-wise product and sum of the embeddings and feeds them to a two-layer perceptron with hidden layer of width $2N$ (as in e.g. (Tai et al., 2015)).[8] For the STS task, we follow this by score regression using class interpolation as in (Tai et al., 2015).

When training for a ranking task (Answer Sentence Selection), we use the bipartite ranking version of Ranknet (Burges et al., 2005) as the objective; when training for STS task, we use Pearson's $r$ formula as the objective; for binary classification

---

[5]In a manner, they resemble tweet data, but without the length restriction and with heavily technical jargon, interspersed command sequences etc.

[6]As in past papers, we use only the first 1M pairs (10%) of the training set.

[7]https://github.com/rkadlec/ ubuntu-ranking-dataset-creator

[8]The motivation is to capture both angle and euclid distance in multiple weighed sums. Past literature uses absolute difference rather than sum, but both performed equally in our experiments and we adopted sum for technical reasons.

| Dataset | Train pairs | Val. pairs | Test pairs | Val.-Test $r$ | Ev. $\#s_0$ | Ev. $\#s_1$ per $s_0$ |
|---|---|---|---|---|---|---|
| `wang` | 44648 | 1149 | 1518 | -0.078 | 178 | 34.9 $\pm 131\%$ |
| `yodaqa/large2470` | 220846 | 55052 | 120069 | 0.348 | 1100 | 159.2 $\pm 100\%$ |
| `wqmprop` | 407465 | 137235 | 277509 | TODO | 3430 | 118.753 $\pm 85\%$ |
| Ubuntu Dialogue v2 | 1M | 195600 | 189200 | 0.884 | 38480 | 10 |

Figure 1: The Val.-Test column shows inter-trial Pearson's $r$ of validation and test MRRs, averaged across the models we benchmarked (see below). The $s_0$ and $s_1$ statistics are shown for the evaluation (Ev. — validation and test) portion of the datasets. The last column includes relative standard deviation of the number of candidate sentences per question, which corresponds to the variation in the difficulty of the ranking task (as well as variation in expected measure values for individual questions).

tasks, we use the binary crossentropy objective.

## 3.1 Baselines

In order to anchor the reported performance, we report several basic methods. **Weighed word overlaps** metrics TF-IDF and BM25 (Robertson et al., 1995) are inspired by IR research and provide strong baselines for many tasks. We treat $s_0$ as the query and $s_1$ as the document, counting the number of common words and weighing them appropriately. IDF is determined on the training set.

The **avg** metric represents the baseline method when using word embeddings that proved successful e.g. in (Yu et al., 2014) or (Weston et al., 2014), simply taking the mean vector of the word embedding sequence and training an $U$ weight matrix $N \times 2N$ that projects both embeddings to the same vector space, $E_i = \tanh(U \cdot \bar{e}_i)$, where the MLP scorer compares them. During training, $p = 1/3$ standard (elementwise) dropout is applied on the input embeddings.

A simple extension of the above are the **DAN** Deep Averaging Networks (Iyyer et al., 2015), which were shown to adequately replace much more complex models in some tasks. Two dense perceptron layers are stacked between the mean and projection, relu is used instead of tanh as the non-linearity, and word-level dropout is used instead of elementwise dropout.

## 3.2 Recurrent Neural Networks

**RNN** with memory units are popular models for processing sentenes (Tan et al., 2015) (Lowe et al., 2015) (Bowman et al., 2015). We use a bidirectional network with $2N$ GRU memory units[9] (Cho et al., 2014) in each direction; the final unit states are summed across the per-direction GRUs

to yield a $2N$ vector representation of the sentence. Like in the avg baseline, a projection matrix is applied on this representation and final vectors compared by an MLP scorer. We have found that applying massive dropout $p = 4/5$ both on the input and output of the network helps to avoid overfitting even early in the training.

## 3.3 Convolutional Neural Networks

**CNN** with sentence-wide pooling layer are also popular models for processing sentences (Yu et al., 2014) (Tan et al., 2015) (Severyn and Moschitti, 2015) (He et al., 2015) (Kadlec et al., 2015). We apply a multi-channel convolution (Kim, 2014) with single-token channel of $N$ convolutions and 2, 3, 4 and 5-token channels of $N/2$ convolutions each, relu transfer function, max-pooling over the whole sentence, and as above a projection to shared space and an MLP scorer. Dropout is not applied.

## 3.4 RNN-CNN Model

The **RNN-CNN** model aims to combine both recurrent and convolutional networks by using the memory unit states in each token as the new representation of the token which is then fed to the convolutional network. Inspired by (Tan et al., 2015), the aim of this model is to allow the RNN to model long-term dependencies and model contextual representations of words, while taking advantage of the CNN and pooling operation for crisp selection of the gist of the sentence. We use the same parameters as for the individual models, but with no dropout and reducing the number of parameters by using only $N$ memory units per direction.

## 3.5 Attention-Based Models

The idea of attention models is to attend preferentially to some parts of the sentence when building its representation (Hermann et al., 2015) (Tan et al., 2015) (dos Santos et al., 2016) (Rocktäschel

---

[9]While the LSTM architecture is more popular, we have found the GRU results are equivalent while the number of parameters is reduced.

et al., 2015). There are many ways to model attention, we adopt the (Tan et al., 2015) model **attn1511** as a conceptually simple and easy to implement baseline. It asymmetrically extends the RNN-CNN model by extra links from $s_0$ CNN output to the post-recurrent representation of each $s_1$ token, determining an attention level for each token by weighed sum of the token vector elements, focusing on the relevant $s_1$ segment by transforming the attention levels using softmax and multiplying the token representations by the attention levels before they are fed to the convolutional network.

Convolutional network weights are not shared between the two sentences and the convolutional network output is not projected before applying the MLP scorer. The CNN used here is single-channel with $2N$ convolution filters 3 tokens wide.

## 4 Model Performance

### 4.1 `dataset-sts` framework

To easily implement models, dataset loaders and task adapters in a modular fashion so that any model can be easily run on any $f_2$-type task, we have created a new software package `dataset-sts` that integrates a variety of datasets, a Python dataset adapter `PySTS` and a Python library for easy construction of deep neural NLP models for semantic sentence pair scoring `KeraSTS` that uses the Keras machine learning library (Chollet, 2015). The framework is available for other researchers as open source on GitHub.[10]

### 4.2 Experimental Setting

We use $N = 300$ dimensional GloVe embeddings matrix pretrained on Wikipedia 2014 + Gigaword 5 (Pennington et al., 2014) that we keep adaptable during training; words in the training set not included in the pretrained model are initialized by random vectors uniformly sampled from $[-0.25, +0.25]$ to match the embedding standard deviation.

Word overlap is an important feature in many $f_2$-type tasks (Yu et al., 2014) (Severyn and Moschitti, 2015), especially when the sentences may contain named entities, numeric or other data for which no embedding is available. As a workaround, ensemble of world overlap count and neural model score is typically used to produce

the final score. In line with this idea, in the Answer Sentence Selection `wang` and `large2470` datasets, we use the BM25 overlap baseline as an additional input to the MLP scoring module, and prune the scored samples to top 20 based on BM25.[11] Furthermore, we extend the embedding of each input token by several extra dimensions carrying boolean flags — bigram overlap, unigram overlap (except stopwords and interpunction), and whether the token starts with a capital letter or is a number.

Particular hyperparameters are tuned primarily on the `yodaqa/large2470` dataset unless noted otherwise in the respective results table caption. We apply $10^{-4}$ $\mathbb{L}_2$ regularization and use Adam optimization with standard parameters (Kingma and Ba, 2014). In the answer selection tasks, we train on 1/4 of the dataset in each epoch. After training, we use the epoch with best validation performance; sadly, we typically observe heavy overfitting as training progresses and rarely use a model from later than a couple of epochs.

### 4.3 Evaluation Methodology

We report model performance averaged across 16 training runs (with different seeds). A consideration we must emphasize is that randomness plays a large role in neural models both in terms of randomized weight initialization and stochastic dropout. For example, the typical methodology for reporting results on the `wang` dataset is to evaluate and report a single test run after tuning on the dev set,[12] but `wang` test MRR has empirical standard deviation of 0.025 across repeated runs of our attn1511 model, which is more than twice the gap between every two successive papers pushing the state-of-art on this dataset! See the *-marked sample in Fig. 2 for a practical example of this phenomenon. Furthermore, on more complex tasks (Answer Sentence Selection in particular, see Fig. 1) the validation set performance is not a great approximator for test set performance and a strategy like picking the training run with best validation performance would lead just to overfitting on the validation set.

To allow comparison between models (and with

---

[10]URL redacted but included as supplementary material

[11]This reduces the number of (massively irrelevant) training samples, but we observed no adverse effects of that, while it speeds up training greatly and models well a typical Information Retrieval scenario where fast pre-scoring of candidates is essential.

[12]Confirmed by personal communication with paper authors.

future models), we therefore report also 95% confidence intervals for each model performance estimate, as determined from the empirical standard deviation using Student's t-distribution.[13]

## 4.4 Results

In Fig. 2 to 4, we show the cross-task performance of our models. We can observe an effect analogous to what has been described in (Kadlec et al., 2015) — when the dataset is smaller, CNN models are preferrable, while larger dataset allows RNN models to capture the text comprehension task better. IR baselines provide strong competition and finding new ways to ensemble them with models should prove beneficial in the future.[14] This is especially apparent in the new Answer Sentence Selection datasets that have very large number of sentence candidates per question. The attention mechanism also has the highest impact in this kind of Information Retrieval task.

While our models clearly yet lag behind the state-of-art on the RTE and STS tasks, it establishes the new baseline on the Ubuntu Dialogue dataset and it is not possible to statistically determine its relation to state-of-art on the `wang` Answer Sentence Selection dataset.

## 5 Model Reusability

To confirm the hypothesis that our models learn a generic task akin to some form of text comprehension, we trained a model on the large Ubuntu Dialogue dataset (Next Utterance Ranking task) and transferred the weights and retrained the model instance on other tasks. We used the RNN model for the experiment in a configuration with dot-product scorer and smaller dimensionality (which works much better on the Ubuntu dataset). This configuration is shown in the respective result tables as **Ubu. RNN** and it consistently ranks as the best or among the best classifiers, dramatically outperforing the baseline RNN model.[15]

During our experiments, we have noticed that it is important not to apply dropout during re-

training if it wasn't applied during the source model training, to balance the dataset labels, and we used the RMSprop training procedure since Adam's learning rate annealing schedule might not be appropriate for weight re-training. We have also tried freezing the weights of some layers, but this never yielded a significant improvement.

## 6 Conclusion

We have unified a variety of tasks in a single scientific framework of sentence pair scoring, and demonstrated a platform for general modelling of this problem and aggregate benchmarking of these models across many datasets. Promising initial transfer learning results suggest that a quest for generic neural model capable of task-independent text comprehension is becoming a meaningful pursuit. The open source nature of our framework and the implementation choice of a popular and extensible deep learning library allows for high reusability of our research and easy extensions with further more advanced models.

Based on our benchmarks, as a primary model for applications on new $f_2$-type tasks, we can recommend either the RNN-CNN model or transfer learning based on the Ubu. RNN model.

## 6.1 Future Work

Due to the very wide scope of the $f_2$-problem scope, we leave some popular tasks and datasets as future work. A popular instance of sentence pair scoring is the question answering task of the **Memory Networks** (supported by the baBi dataset) (Weston et al., 2015). A realistic large question **Paraphrasing** dataset based on the AskUbuntu Stack Overflow forum had been recently proposed (Lei et al., 2015).[16] In a multi-lingual context, sentence-level MT Quality Estimation is a meta-task with several available datasets.[17] While the tasks of **Semantic Textual Similarity** (supported by a dataset from the STS track of the SemEval conferences (Agirre et al., 2015)) and **Paraphrasing** (based on the Microsoft Research Paraphrase Corpus (Dolan and Brockett, 2005) right now) are available within our framework, we do not report the results here as the models lag behind the state-of-art significantly and

---

[13]Over larger number of samples, this estimate converges to the normal distribution confidence levels. Note that the confidence interval determines the range of the true expected evaluation, not evaluation of any measured sample.

[14]We have tried simple averaging of predictions (as per (Kadlec et al., 2015)), but the benefit was small and inconsistent.

[15]The RNN configuration used for the transfer, when trained only on the target task, is not shown in the tables but has always been worse than the baseline RNN configuration.

[16]The task resembles paraphrasing, but is evaluated as an Information Retrieval task much closer to Answer Sentence Selection.

[17]`http://www.statmt.org/wmt15/quality-estimation-task.html`

| Model | wang MAP | wang MRR | l2470 MAP | l2470 MRR | wqm MAP | wqm MRR |
|---|---|---|---|---|---|---|
| (Tan et al., 2015) | 0.728 | 0.832 | | | | |
| (dos Santos et al., 2016) | 0.753 | 0.851 | | | | |
| TF-IDF | 0.578 | 0.709 | 0.267 | 0.363 | | |
| BM25 | 0.630 | 0.765 | 0.314 | 0.491 | | |
| RNN w/o BM25 | 0.649 ±0.011 | 0.743 ±0.010 | 0.262 ±0.003 | 0.381 ±0.008 | | |
| avg | 0.713 ±0.003 | 0.806 ±0.005 | 0.278 ±0.003 | 0.481 ±0.008 | | |
| DAN | 0.709 ±0.004 | 0.787 ±0.007 | 0.282 ±0.004 | 0.490 ±0.010 | | |
| RNN | 0.696 ±0.006 | 0.785 ±0.007 | 0.277 ±0.004 | 0.487 ±0.008 | | |
| CNN | 0.717 ±0.005 | 0.793 ±0.005 | 0.288 ±0.003 | 0.499 ±0.007 | | |
| RNN-CNN | 0.729 ±0.006 | 0.810 ±0.009 | 0.288 ±0.004 | 0.503 ±0.010 | | |
| attn1511 | 0.732 ±0.006 | 0.817 ±0.012 | 0.286 ±0.003 | 0.499 ±0.009 | | |
| *attn1511 | 0.756 | 0.859 | | | | |
| Ubu. RNN w/o BM25 | 0.731 ±0.007 | 0.814 ±0.008 | 0.359 ±0.003 | 0.539 ±0.006 | | |
| Ubu. RNN | | | 0.291 ±0.002 | 0.515 ±0.004 | | |

**Figure 2:** Model results on the Answer Sentence Selection task, as measured on the `wang`, `yodaqa/large2470` and `wqmprop` datasets.
* Demonstration of the problematic single-measurement result reporting in past literature — an outlier sample in our 16-trial attn1511 benchmark that would score as a state of art; in total, three outliers in the trial (12.5%) scored better than (Tan et al., 2015).

show little difference in results. Advancing the models to be competitive remains future work. A generalization of our proposed architecture could be applied to the **Hypothesis Evidencing** task of binary classification of a hypothesis sentence $s_0$ based on a number of memory sentences $s_1$, for example within the MCText (Richardson et al., 2013) dataset. We also did not include several major classes of models in our initial evaluation. Most notably, this includes serial RNNs with attention as used e.g. for the RTE task (Rocktäschel et al., 2015), and the skip-thoughts method of sentence embedding. (Kiros et al., 2015)

We believe that the Ubuntu Dialogue Dataset results demonstrate that the time is ripe to push the research models further towards the real-world by allowing for wider sentence variability and less explicit supervision. But in particular, we believe that new models should be developed and tested on tasks with long sentences and wide vocabulary. In terms of models, recent work in many NLP domains (dos Santos et al., 2016) (Cheng et al., 2016) (Kumar et al., 2015) clearly points towards various forms of attention modelling to remove the bottleneck of having to compress the full spectrum of semantics into a single vector of fixed dimensionality. In this paper, we have shown the benefit of training a model on a single dataset and then applying it on another dataset. One open question is whether we could jointly train a model on multiple tasks simultaneously (even if they do not share some output layers). Another option would be to include extra supervision similar to the token overlap features that we already employ; for example, in the new Answer Sentence Selection task datasets, we can explicitly mark the actual tokens representing the answer.

## Acknowledgments

redacted

| Model | MRR | 1-2 R@1 | 1-10 R@1 | 1-10 R@2 | 1-10 R@5 |
|---|---|---|---|---|---|
| * TF-IDF | | 0.749 | 0.488 | 0.587 | 0.763 |
| * RNN | | 0.777 | 0.379 | 0.561 | 0.836 |
| * LSTM | | 0.869 | 0.552 | 0.721 | 0.924 |
| avg | 0.624 | 0.793 | 0.472 | 0.608 | 0.836 |
| | ±0.002 | ±0.002 | ±0.002 | ±0.002 | ±0.003 |
| DAN | 0.578 | 0.792 | 0.493 | 0.615 | 0.830 |
| | ±0.070 | ±0.035 | ±0.074 | ±0.059 | ±0.033 |
| RNN | 0.781 | 0.907 | 0.664 | 0.799 | 0.951 |
| | ±0.003 | ±0.002 | ±0.004 | ±0.004 | ±0.001 |
| CNN | 0.718 | 0.863 | 0.587 | 0.721 | 0.907 |
| | ±0.003 | ±0.002 | ±0.004 | ±0.005 | ±0.003 |
| RNN-CNN | **0.788** | **0.911** | **0.672** | **0.809** | **0.956** |
| | ±0.001 | ±0.001 | ±0.002 | ±0.002 | ±0.001 |
| attn1511 | 0.772 | 0.903 | 0.653 | 0.788 | 0.945 |
| | ±0.004 | ±0.002 | ±0.005 | ±0.005 | ±0.002 |

Figure 3: Model results on the Ubuntu Dialogue next utterance ranking task. Models use slightly specific configuration due to much bigger dataset (in terms of both samples and sentence lengths) — only 160 tokens are considered per input, no dropout is applied, RNN use $N$ memory units, projection matrix is only $N \times N$ and the dot-product scorer is used for comparison. The attn1511 model furthermore has only $N/2$ RNN memory units and $N/2$ CNN filters.
* Exact models from (Lowe et al., 2015) reran on the v2 version of the dataset (by personal communication with Ryan Lowe) — note that the results in (Lowe et al., 2015) and (Kadlec et al., 2015) are on dataset v1 and not directly comparable.

| Model | SICK-2014 STS $r$ test | SICK-2014 3-RTE train | SICK-2014 3-RTE test | SNLI 3-RTE train | SNLI 3-RTE test |
|---|---|---|---|---|---|
| (Tai et al., 2015) | 0.868 | | | | |
| (Bowman et al., 2015) LSTM | | 1.000 | 0.713 | 0.848 | 0.776 |
| (Bowman et al., 2015) Tran. | | 0.999 | 0.808 | | |
| (Cheng et al., 2016) | | | | 0.921 | 0.890 |
| TF-IDF | 0.479 | | | | |
| BM25 | 0.474 | | | | |
| avg | 0.621 | 0.770 | 0.652 | 0.735 | 0.710 |
| | ±0.017 | ±0.020 | ±0.017 | ±0.014 | ±0.008 |
| DAN | 0.642 | 0.715 | 0.662 | 0.718 | 0.708 |
| | ±0.016 | ±0.010 | ±0.003 | ±0.009 | ±0.002 |
| RNN | 0.664 | 0.759 | 0.732 | 0.784 | 0.749 |
| | ±0.022 | ±0.016 | ±0.010 | ±0.019 | ±0.010 |
| CNN | 0.762 | 0.927 | 0.799 | | |
| | ±0.006 | ±0.008 | ±0.004 | | |
| RNN-CNN | 0.790 | 0.765 | 0.709 | 0.811 | 0.753 |
| | ±0.005 | ±0.084 | ±0.059 | ±0.037 | ±0.008 |
| attn1511 | 0.723 | 0.858 | 0.767 | 0.829 | 0.774 |
| | ±0.009 | ±0.010 | ±0.004 | ±0.014 | ±0.004 |
| Ubu. RNN | 0.799 | 0.931 | 0.813 | | |
| | ±0.009 | ±0.017 | ±0.005 | | |
| SNLI RNN | 0.798 | 0.927 | 0.831 | | |
| | ±0.007 | ±0.006 | ±0.002 | | |

Figure 4: Model results on the STS and RTE tasks, reporting Pearson's $r$ and 3-class accuracy, respectively.

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
