# Peer review of "Sentence Pair Scoring: Towards Unified Framework for Text Comprehension"

_CoNLL 2016 — decision unknown_

[Official Review · Reviewer 1 · rating 4 · confidence 2]
soundness 4 · originality 4 · clarity 4 · impact 4 · substance 3 · appropriateness 5 · meaningful comparison 4 · replicability 4 · presentation format Poster

This paper proposes the new (to my knowledge) step of proposing to treat a
number of sentence pair scoring tasks (e.g. Answer Set Scoring, RTE,
Paraphrasing,
among others) as instances of a more general task of understanding semantic
relations
between two sentences. Furthermore, they investigate the potential of learning
generally-
applicable neural network models for the family of tasks. I find this to be an
exciting
proposal that's worthy of both presentation at CoNLL and further discussion and
investigation.

The main problem I have with the paper is that it in fact feels unfinished. It
should be
accepted for publication only with the proviso that a number of updates will be
made
for the final version:
1 - the first results table needs to be completed
2 - given the large number of individual results, the written discussion of
results
is terribly short. Much more interpretation and discussion of the results is
sorely needed.
3 - the abstract promises presentation of a new, more challenging dataset which
the paper
does not seem to deliver. This incongruity needs to be resolved.
4 - the results vary quite a bit across different tasks - could some
investigation be made into
how and why the models fail for some of the tasks, and how and why they succeed
for others?
Even if no solid answer is found, it would be interesting to hear the authors'
position regarding
whether this is a question of modeling or rather dissimilarity between the
tasks. Does it really
work to group them into a unified whole?
5 - please include example instances of the various datasets used, including
both prototypical
sentence pairs and pairs which pose problems for classification
6 - the Ubu. RNN transfer learning model is recommended for new tasks, but is
this because
of the nature of the data (is it a more general task) or rather the size of the
dataset? How can
we determine an answer to that question?

Despite the unpolished nature of the paper, though, it's an exciting approach
that
could generate much interesting discussion, and I'd be happy to see it
published
IN A MORE FINISHED FORM.
I do recognize that this view may not be shared by other reviewers!

Some minor points about language:
* "weigh" and "weighed" are consistently used in contexts that rather require
"weight" and
"weighted"
* there are several misspellings of "sentence" (as "sentene")
* what is "interpunction"?
* one instance of "world overlap" instead of "word overlap"